# The Consequence of Combining Indigenous Techniques with a Flexible Design to Reduce Energy Consumption in Residential Buildings for Future Architecture

**Hoda Ramezani ***[ID] **and Ehsan Reza** [ID]

Faculty of Fine Arts, Design and Architecture, Department of Architecture, Cyprus International University, Haspolat-Lefkoşa, Mersin 10, Nicosia 99258, Turkey
* Correspondence: hodaramezani127@gmail.com

**Abstract:** A significant amount of research has addressed the issue of energy consumption reduction and the use of natural ventilation. Nevertheless, fewer studies have discussed the use of vernacular solutions and their integration with modern architecture on a global scale. Therefore, the primary motivation of this article is to answer the question of how combining indigenous techniques for natural ventilation with characteristics of flexible design can be reflected in reducing energy consumption in residential buildings Since natural ventilation is one of the most effective factors in creating thermal comfort and this factor creates comfortable conditions in hot and humid climates by taking advantage of airflow, reducing relative humidity, and increasing surface evaporation, this article examines the Shavadoon, which is an indigenous technique to escape from the excruciating heat of summer in Iran, and, by integrating it with modern architecture in Northern Cyprus, it seeks its effects on cooling the building and reducing energy consumption. The details of local solutions, natural ventilation, flexible design, and modern architecture will be extracted from the literature. Subsequently, via case study and, eventually, using the knowledge of BIM, the effect of their integration in reducing energy consumption will be investigated. Finally, new solutions for future architects in line with the construction of energy-efficient residential buildings will be provided.

**Keywords:** reducing energy consumption; adapting architecture; flexible design; sustainability; building information modeling (BIM); Shavadoon

## 1. Introduction

Rapid technology improvement, construction, and growing urbanization are some of the major factors that have caused problems for nature exploitation, which is irreparably demolished nature. Moreover, environmental contamination is equally leading to a reduction in energy sources [1].

According to the IPCC statement (Intergovernmental Panel on Climate Change), the developing area is the main reason for 32% of the world's final energy usage, 19% of greenhouse gases, which are related to energy consumption (GHG) emissions, and 51% of global electricity [2].

The increasing need for electricity and gas within residential buildings for both cooling and heating (air conditioning), doubt in the energy market in the last 30 years, and people's alertness regarding the effect of the usage of fossils fuels in the environment are the key pressures behind improving research concentration to discover ways for decreasing residential buildings trail [3].

Furthermore, cooling and heating, air conditioning, and air conditioning energy needs cover about 34.8% of the energy consumption of buildings in the U.S [4]. The combination of both high temperature and humidity can directly increase the energy needed to balance and control the weather [5]. For example, a great level of energy for air conditioning usage in hot and humid summers, electricity energy consumption, as well as gas usage for

warming the place in winter in both commercial and residential constructions in Florida are estimated to be around 23.7% and 28.6% of the growing energy requirements [6].

Additionally, recent research results illustrated that, due to the progressive growth in regular temperatures in 2050 and 2080, the heat energy request for heating will reduce, while the requirement of electricity for cooling will increase over this term [7].

So, any kind of improvement in building energy efficiency can have a significant role in the general energy countries' expenses. Furthermore, a reduction in energy consumption can play an important role for humans from a financial and well-being point of view and also has an improving effect on environmental sustainability [8].

In the report of the European Energy Performance of Buildings Directive (EPBD) conducted in 2003, there are three sources for modifying the energy efficiency of residential buildings: (1) mitigating climate change; (2) ensuring energy security; and (3) eradicating energy poverty [9].

Since previous construction was associated with preserving the environment and vernacular solutions meet the requirements of residents, the study on the integration and use of indigenous solutions with the modern buildings and its effect on reducing energy consumption has the ability to turn it into an attractive topic.

Moreover, with the improvement and increase in human awareness about preserving the environment and replacing renewable energy sources, there has been development in the design and construction industry. Today, BIM has changed the way projects are presented in various industries. Intelligence and efficiency add to project execution and connect teams, data, and workflows at every stage of a project in the cloud for better project outcomes (Figure 1).

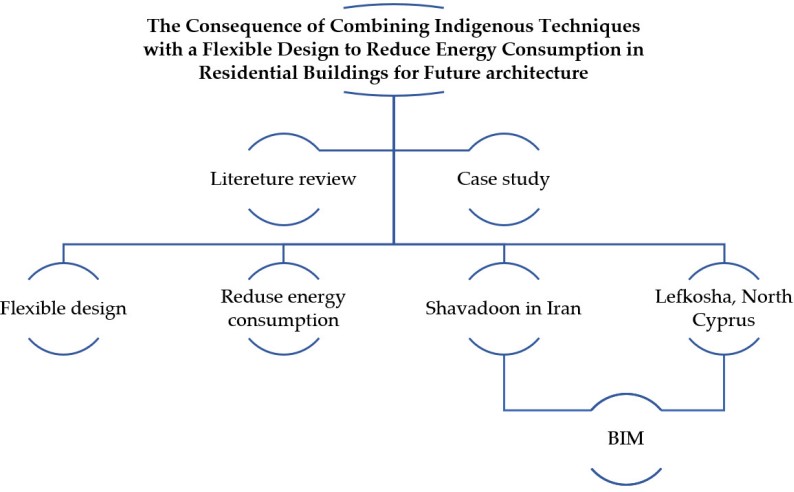

**Figure 1.** An Overview of the Research Proses (by authors).

Hence, this paper tries to answer these questions in two parts; first, using the literature review approach, and, second, through case study analysis.

1. What is the relationship between indigenous techniques and energy efficiency?

2. What are the parameters that can be changed and optimized in the architectural design phase of buildings?

3. How can the integration of vernacular techniques and modern architecture have an effect on the sustainability of future housing?

## 2. Materials and Methods

Sustainable architecture, which is actually a subset of sustainable design, can perhaps be considered as one of the important contemporary topics, which is a logical reaction to the problems of the industrial age. Possible strategies to increase the sustainability of residential buildings include considering the footprint of the building and surrounding environment, the materials, as well as the design considerations that are passive.

This information requires the use of modern science. BIM stands for Building Information Model, which is considered as one of the latest and newest technologies that is widely used today.

BIM is a method of examining and studying different parts of the building step by step, which helps in completing the building process, measuring, drawing maps, and adding various parameters during the project, along with many other tasks.

Qualitative methods such as grounded theory have been used for the literature review, and BIM has been used to analyze thermal comfort and the effects of ventilation on building cooling. The proposed framework includes (1) the correlation between natural ventilation, vernacular techniques, and flexible design and (2) the creation of the BIM model because it shows the geometry of the building with all the details of the building.

It also shows the properties of materials and property information such as thermal conductivity and solar transmission. Subsequently, the results were interpreted and analyzed, which will be discussed in detail in the subsequent chapters.

## 3. Alignment of Vernacular Architecture and Flexible Design to Reduce Energy Consumption

Within years, vernacular residences have been reclaimed and have therefore remained as cultures and societal evidence providing a straight connection to the past [10].

Due to the bioclimatic features that have been integrated into the native architecture, the protection of vernacular dwellings is considered a sustainable approach [11].

The maintenance and energy advancement of local residences of different importance over the previous and future times is the foundation of meaning and importance [12].

In addition, retrofitting is a recent term which is nowadays used for explaining the procedure of restoring and remodeling buildings with the intention of improving the buildings' energy efficiency and, therefore, reducing energy usage [13].

It is valorous to perceive that the phrase retrofitting is used in a diverse way compared to the sense of conservation. Whilst conservation mostly adverts to the preservation of the aesthetic and epochal worthiness of a dwelling, retrofitting gives particular emphasis to energy diminution [14].

On the other hand, the terms of flexibility will also try to help in reducing the energy consumption. The idea of the "flexible design" of residential units from the beginning of the 20th century as part of the modern movement was introduced by architects such as "Le Corbusier", "Miss Vanderrohe", and, subsequently, "Habraken" and "Hertz Berger" [15]. This idea was due to technological developments and events in the period that made it possible to separate the fixed structures of the building from its elements [14]. In the design process, flexibility includes activities related to variability to achieve new performance and usability. In architecture and environmental design, flexibility means a man-made spatial organization and changes in it to achieve new conditions, needs, and applications [16].

Adaptable space is a decisive response to the efforts of modern architects to create flexibility, which has multiple goals and is not created for a single specific purpose. Flexible buildings seek to respond to changing conditions of use, performance, or location. Buildings have a long and complex life cycle, during which their functional parameters change extensively [15].

The flexible system is defined with the aim of obtaining appropriate methods for creating a kind of dimensional coordination between the components of the building. The utilization of modulated and coordinated dimensions in design and execution allows for the production of buildings in an industrial and prefabricated manner and in a mass and open system [14].

The flexible system plays an important role in the design, construction, and installation of the building and affects two basic factors, namely, dimensions and position, and reduces the waste of materials and consumables during production and changes in parts. This is very important because, especially in industrial production and mass production, it causes the materials to be used properly and prevents their unnecessary waste [17].

New design theory ensures flexibility in various fields such as the use of space, reducing energy consumption, and the use of ventilation systems, allowing for improvement, while conventional building systems are not designed for change, and any deformation in the building will be accompanied by the destruction of part or sometimes all of it [15].

### 3.1. Vernacular Architectural Solutions to Reduce Energy Consumption

In vernacular architecture, buildings are unified against their surroundings to create a natural harmony between the climate, architecture, and people. Indigenous architecture has been built on people's experiences, which were environment- and climate-friendly [18].

The expansion of vernacular architecture is focused on the functions of the building, which change over time depending on various factors such as weather (the amount of sunlight, humidity, rain, etc.), environment (forest, mountain, or near the sea), culture (family size, how to use the building, social conditions, etc.), and economic conditions, which are modified and evolve [19].

Therefore, it can be mentioned that this type of architecture, in addition to using local materials and having a great effect on the sustainability of the building, has many other advantages, including the use of native knowledge, creating a vital connection between humans and the living or working environment, and they can be in harmony with the weather conditions and create a positive performance in the use of the building [20].

For example, ancient Egyptian miners in the excavation of a mine faced the issue of not having enough air ventilation, which caused suffocation and death, which was common due to a lack of air. They solved the problem by adding some shaft openings with crosscuts to make air move down one shaft and up the next. Natural ventilation occurs in Egyptian underground excavations because the outside air temperature lowers so substantially at night that a cool air draught flows into the chamber, flushing out the stale air and dust and replacing it with fresh air [21].

A long-term technique to improve energy efficiency and minimize carbon emissions in buildings is natural ventilation (NV), which is also the best alternative strategy for reducing the consumption of fossil fuels [22].

In natural ventilation, the temperature difference will cause air circulation that uses the natural forces of wind and buoyancy to bring fresh air into the interior of buildings and distribute it properly to optimize thermal comfort. It can provide sufficient breathing air, suitable pollutant ventilation, and suitable temperature ventilation [23].

### 3.2. Natural Ventilation in Vernacular Architecture to Consumption Energy

Better thermal comfort in the interior space of native architecture has been based on the integration of passive design, which has led to a reduction in energy consumption [24]. According to these strategies, to provide more thermal comfort in the hot season, it is better to pay attention to the proper shading of the openings and the building cover, natural ventilation, suitable undergrowth, and the high current figure of the building envelope [10].

It has also been emphasized that, in a moderate climate with a high external humidity, it is possible to provide adequate thermal comfort with ventilation during the day [25].

For this purpose, the best result occurs when the outside air temperature is around 28 and 32 °C and the speediness inside is between 1.5 and 2 m/s. At night, if the air outside the house is 20 °C and the daily fluctuation is 10 °C, it is considered a successful strategy if the thermal conditions inside the building decrease during the next day [12].

In general, night ventilation reduces the indoor space's air temperature and at the same time raises the time among the maximum outdoor and indoor temperatures [25].

Some of the field studies that have been conducted on the Eastern Mediterranean vernacular architecture prove that the current mass efficiency of the traditional building covers is connected to the construction resources and openings' small size. Precisely, the studies show that thermal mass influence aims to achieve an improvement in indoor temperature constancy along with a raise in a gap within indoor and outdoor highest temperatures. The application of ventilation during the night by means of an inert design

technique in Cyprus urban vernacular architecture in its hot summer time is suggested by Philokyprou and Michael, 2021 [10].

Due to the claims of the same researchers, ventilation in a day throughout the summer times is not wanted due to high outdoor temperatures. Furthermore, it is mentioned that natural ventilation is not demanded in the wintertime, as the external temperature is less than the desired comfort zone temperature [26].

The results of a recent training on nighttime ventilation in Cyprus vernacular architecture specify that cross ventilation ensures thermal comfort inside buildings in the summertime, while single-sided ventilation can be appropriate in the intermediate period [13].

## 4. Indigenous Techniques (Shavadoon)—The Procedure of Heating and Cooling in the Building

Shavadoon is an underground space that has been used as a native technique by using the average temperature of the earth to create thermal comfort for residents in the hot and semi-humid city of Dezful, located in Iran [27]. This system engages the high land thermal valency; two have temperatures around 20 centigrade below the surrounding temperature on summer's hottest days, and they are constructed 5–12 m deep [28]. This place, in the past, has been used as a daily resting place as well as for food storage and general refrigeration needs. Shavadoons are combined of various components such as an entrance that is rather wide and is located in a part of the courtyard [29].

At the stairway at the entrance to the Sahn of Shavadoon, footrests are created after some stairs, like the landing space in today's buildings. The main hall (Sahn) is the main element that is the center of activities. The Kat is comprised of some small rooms that are separated from the court by a maximum one-level difference. Additionally, in some kinds of Shavadoon, those rooms (Kat) have access to neighboring Kat spaces with Tal and Darizeh (vertical canals), which are embedded for light and ventilation [27] (see Figure 2).

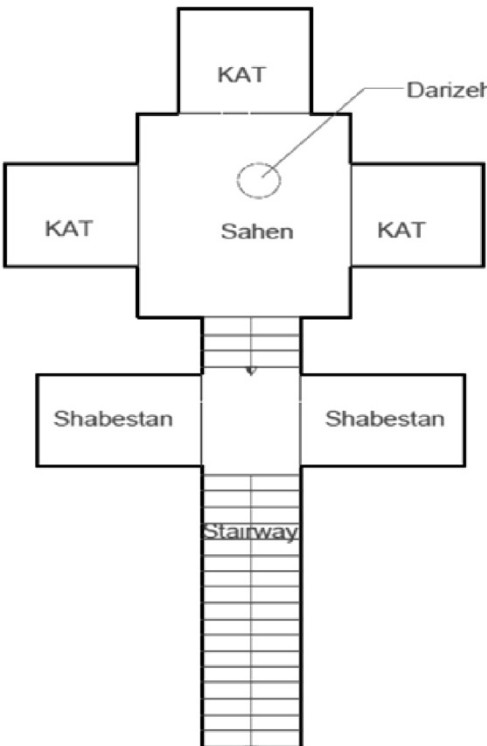

**Figure 2.** Original form of the Shavadoon plan [27].

The air temperature in Shavadoon hinges on two factors: the ground's crust temperature and the quantity of natural ventilation. The earth's crust is separated into two parts. The primitive sector is the above part, which is immediately in contact with the

surroundings and is afoul of a fast temperature conversion because of the routine daily and seasonal temperature changes. The other part is under the first part and spreads to depths of 1–20 m. Examining the differences in land temperature at various depths throughout the year is important because temperature changes affect heat exchanges between underground buildings and the rest of the environment [22].

Examinations in this regard shows that, in summer's hottest days, the Shavadoon temperature is just 25 centigrade, though the environment temperature in the same period of time is more than 45 centigrade. It lets the users stay away from the scorching heat during the day. By simulating Shavadoon for seasons of hot and cold weather, the outcomes reveal that the appropriate use of natural and physical spectacles, such as natural convection, has the potential for providing users with free cooling and heating. The mathematical outcomes revealed a 10% difference from trial outcomes [28]. The performance outcome revealed that the ventilation would be raise by 57% with the help of connecting the Shavadoon by Tals, and using a window can reduce the inlet stream to the Shavadoon.

In natural ventilation inside of the building, it is better to have two-way ventilation, and the airflow enters from one side and exits from the other side of the ceiling of the room [29].

The main factors in improving the temperature of Shavadoon are air conditioning and the use of a constant ground temperature. One aspect of ground ventilation is the use of natural stimuli; this is done unilaterally, and cross-flow ventilation can be achieved [21].

In order to create airflow and take advantage of natural ventilation, holes should be created in the space of the Shavadoon for the entry and exit of air. Surely, it is almost impossible to find cold air in a hot and humid climate; the only way to access cool air is to use the air source in the sky. The farther from the surface of the earth, the cooler the air becomes. The heating of the air in the lower levels is one of the factors in increasing the air pressure that produces vertical currents [26].

During the days, due to the presence of the heated airflow of the earth's surface upwards, it is not possible for the upper cold air to reach the earth. However, at night, when the sun's radiant energy does not reach the ground and the above flow is interrupted, it is a good opportunity for the heavy and cold air of the sky to move down. This airflow first cools the roof and then penetrates into the rest of the space. When it penetrates into the underground spaces, the above air is stored there so that residents can use it during the day [29]. Shavadoon can be ventilated directly by cooling the building during the night. The entrance and openings of the Shavadon should be installed in such a way that the ability to quickly ventilate at night is possible and prevents the air from entering during the day. It does not fall, and the upward movement of warm air also intensifies this [22] (Figure 3).

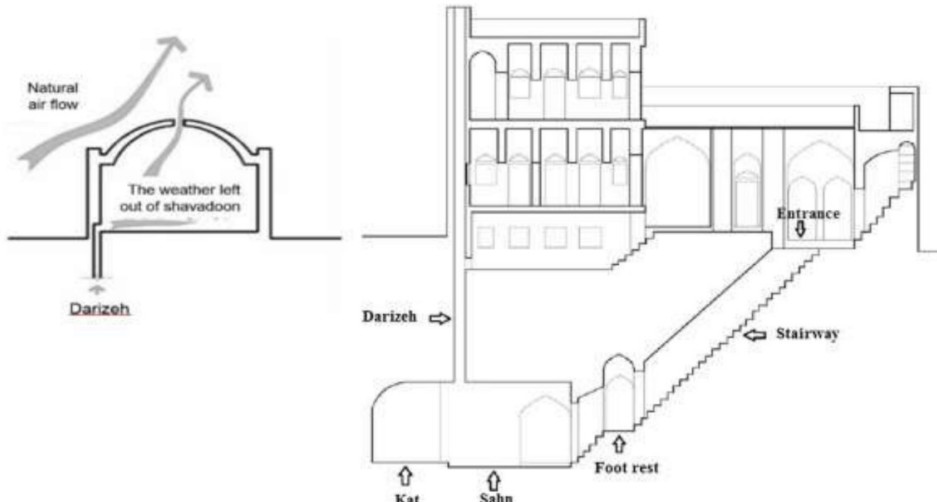

**Figure 3.** Shavadoon Ventilation at Night [22,27,29].

### 5. Integration of the Shavadoon Space and Modern Architecture in North Cyprus from a Flexible Architecture Design Aspect (Case Study)

Dezful is one of the cities of Khuzestan province, which is located in southwest Iran and is located at the latitude of 32 degrees 22 min north and 48 degrees 24 min east (Figure 4a). This city has a mild and semi-humid climate, and on some summer days, the temperature reaches 52 degrees Celsius (Figure 4b) [28].

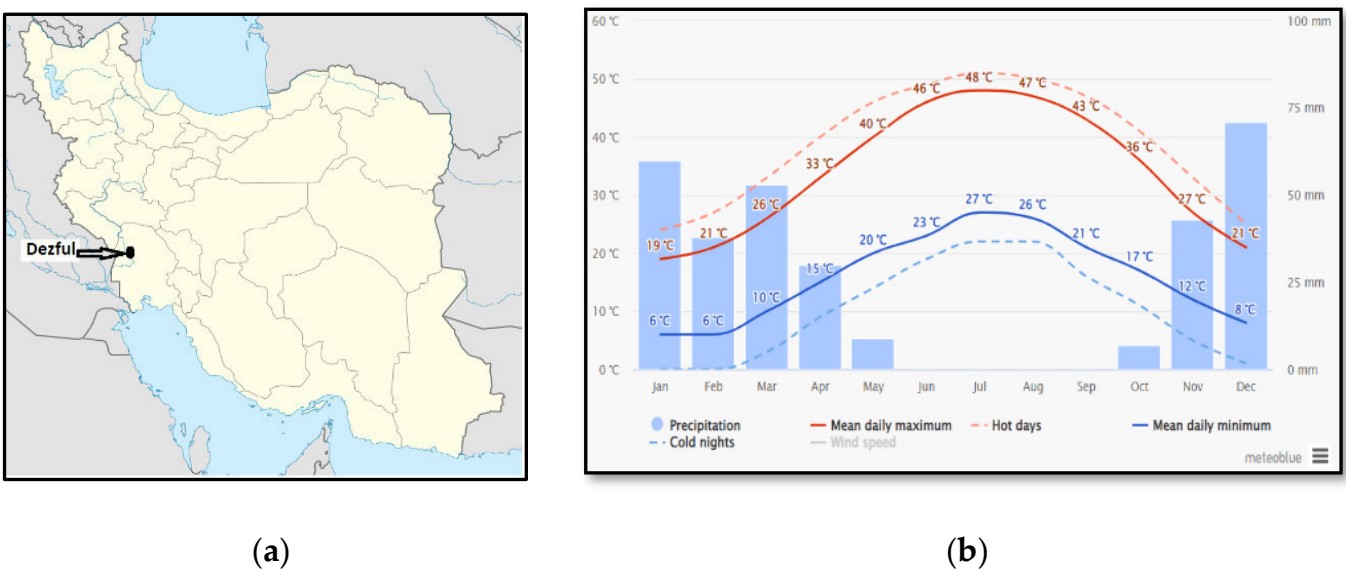

(**a**)          (**b**)

**Figure 4.** (**a**) Location of Dezful in Iran. [30]. (**b**) Average temperatures and precipitation of Dezful [28].

Similar weather is in North Cyprus in the city of Nicosia, with a latitude of 35°10′0.0012″ and a longitude of 33°22′0.0012″ (Figure 5a) [31]. The average annual temperature is 64.0 °F (17.8 °C), and the hottest month, on average, is July, with an average temperature of 82.0 °F (27.8 °C). The coolest month is January, with an average temperature of 48.0 °F (8.9 °C) (Figure 5b) [6,31].

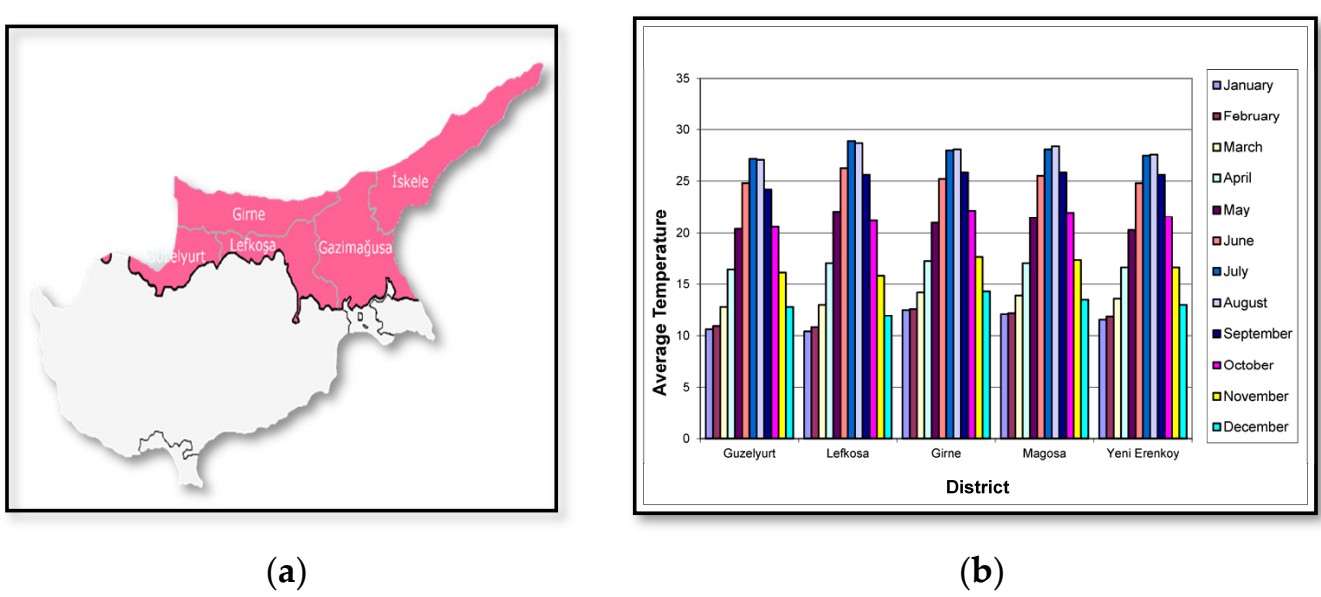

(**a**)          (**b**)

**Figure 5.** (**a**) Lefkoşa in North Cyprus [6]. (**b**) Average temperatures and precipitation of Lefkoşa in North Cyprus [31].

It has been mentioned, in the city of Dezful in Iran, that, by using the force of the earth and by using the Shavadoon, which is underground in the houses, and its holes on the ground, it causes the airflow in a natural way, and, as a result, a decrease of at least 10 degrees has been measured [28]. The shape of this structure was close to that of a normal house, which has a living room and a bedroom, and people used these spaces to have fun during the summer period [32].

Since the lifestyle has changed, and based on population growth, the lack of land, and economic effects, people have turned to living in apartments, which has had an adverse effect on energy consumption, especially with fossil fuel sources [27].

Therefore, by integrating the traditional space into today's apartments, in addition to the effects of using less materials, those space can reduce energy consumption [33].

Regarding the polyvalence term as a feature of the static form in flexible design, which was introduced into the architectural discussion by Hertzberger, it means that the building can be used differently without adjusting for how it is constructed. Polyvalence in the field of housing is primarily related to the ability to exchange activities between different rooms, which can be used in a variety of functions without making a fundamental change [20].

Consequently, compliant with the common principles of flexibility in buildings, the flexible design policies distribute to four tendencies conferring to the practical indecision of the environment. Based on Table 1, "Spatial flexibility in a fixed surface area" is the opportunity of changing an inner space without varying its whole capacity, which requires practical zones and versatile spaces. "Evolutionary spatial flexibility" is the valence of space for improvement and convention and is thus attractive to its lifecycle. "Technological flexibility related to construction techniques" comprises the replacement of irreplaceable components to authorize their adjustment and replacement. "Technological flexibility related to the easy preservation of the connections and building sub-systems," the previous trend, is the usage of intragenic components for easy conservation [14].

**Table 1.** General principles of flexibility in buildings [14].

| Trends | Strategies |
|---|---|
| **Spatial flexibility in a fixed surface area** | Access to redundancy (two or more access points) Personalizes privacy and social needs. Make use of mobile equipment (equip walls, cabinets, or prefabricated modular interior partitions) |
| **Evolutionary spatial flexibility** | Increase the surface area within the confines of the existing support (closure of spaces that are already built) Increase the Dwelling's surface area Add living units to increase the internal surface area. |
| **Technological flexibility related to construction techniques** | The building's adjustability and adaptability Make use of dried and stratified closures. Regularity of structure and adaptability |
| **Technological flexibility related to easy integration of automated home systems maintenance of the installation and building sub-system** | Redundancy and inspection of the equipment |

Accordingly, it can be claimed that the Shavadoon spaces based on general principles of flexibility in buildings can be categorized as "Spatial flexibility in a fixed surface area", which, by some changes in interior design without conversing with the general mass, can be used as a polyvalence space in modern buildings. In addition to affecting the efficiency of the quality of natural ventilation in modern buildings, it also reduces energy consumption, which will help a future building to be more sustainable.

*5.1. Simulation of a Shevadon Space and Its Integration into Modern Architecture in North Cyprus (Software Study)*

Due to the recent attention to reducing waste, greenhouse gas emissions, and energy consumption in the field of sustainability, technologies will help industries through digital assistance [33].

In recent decades, BIM technology has provided essential data and information for projects and methods of analyzing building performance, materials and resources, energy and sustainable sites, internal quality, and more about energy consumption in the direction of sustainable construction [34].

In this regard, different software can be used [35], but BIM as a technology and methodology provides the integration of existing information and promotes the environment of the participation of factors influencing the reduction of waste and energy consumption [36].

Therefore, by using the available tools and knowledge, it has been attempted to add the indigenous technique of Shavadoon in the warm and semi-humid climate in Dezful, which is a city in Iran, and this will be applied in almost the same climate to a modern building in Lefkoşa, which is the capital city of North Cyprus. Even though the weather conditions affect the quality of natural ventilation, and because the Revit software will apply the existing weather conditions on the building using the location, this software was used to model an apartment in Lefkoşa (the capital of Northern Cyprus).

The building is a seven-story apartment with one bedroom and two bedrooms on each floor (see Figure 6).

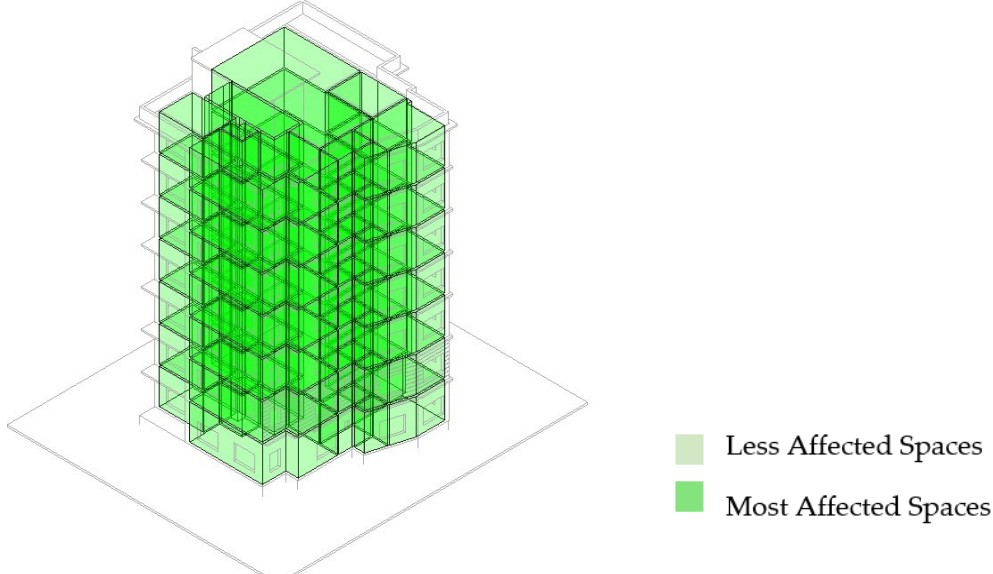

**Figure 6.** Modeling of a Real Building in Lefkoşa (by authors, 2022).

In the first part, the whole building was analyzed and examined using the Inside 360 plugin, which is designed based on ASHRAE Standards. This showed that the building is economical, which, in terms of cooling the residential building in question, is at least 13.1 kW and at most 56.7 kW, which shows that the building is economical for cooling. On the other hand, the peak heating load is 27.9, and the noteworthy point is that, with population growth, construction development, air pollution, and climate change, the weather is becoming hotter, and natural ventilation without filters will endanger the health of residents. Therefore, in the next step, by applying the Shavadon space, we will try to check its effects on the cooling of the building and the reduction in energy consumption (Figure 7).

In the next step, as shown in Figure 8, the Shavadoon, which is used in Iranian traditional residential buildings, is applied to a modern residential building in Lefkoşa by Revit software.

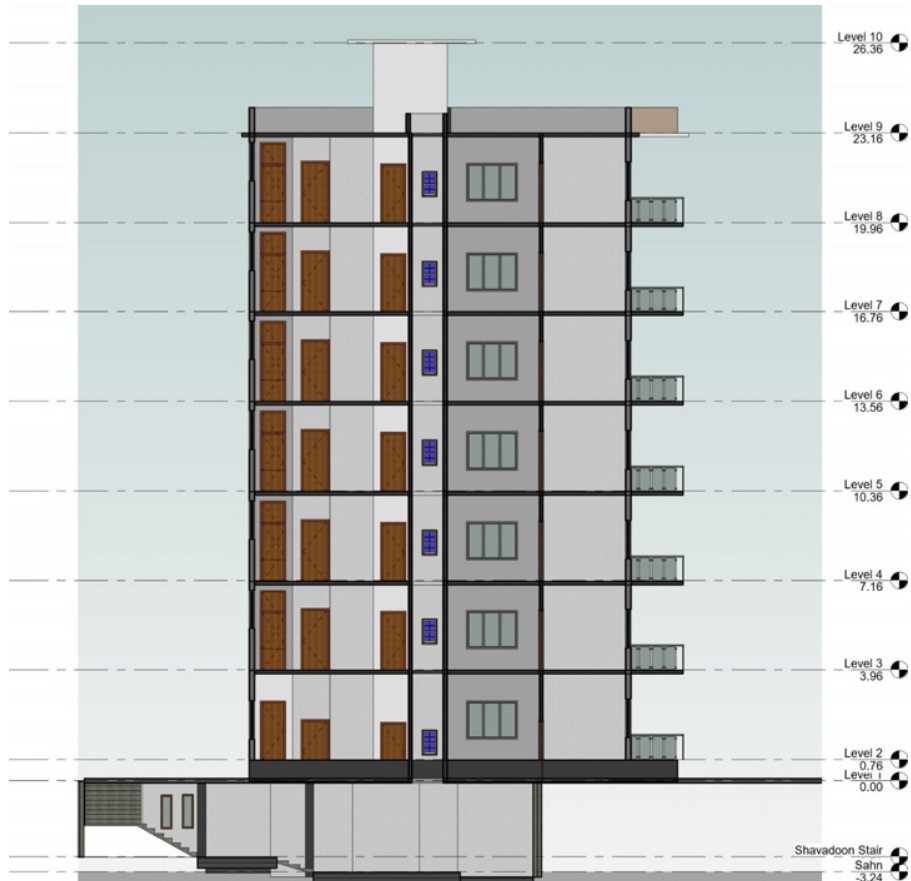

**Figure 7.** Energy Consumption in the Modern Residential Building in Lefkoşa.

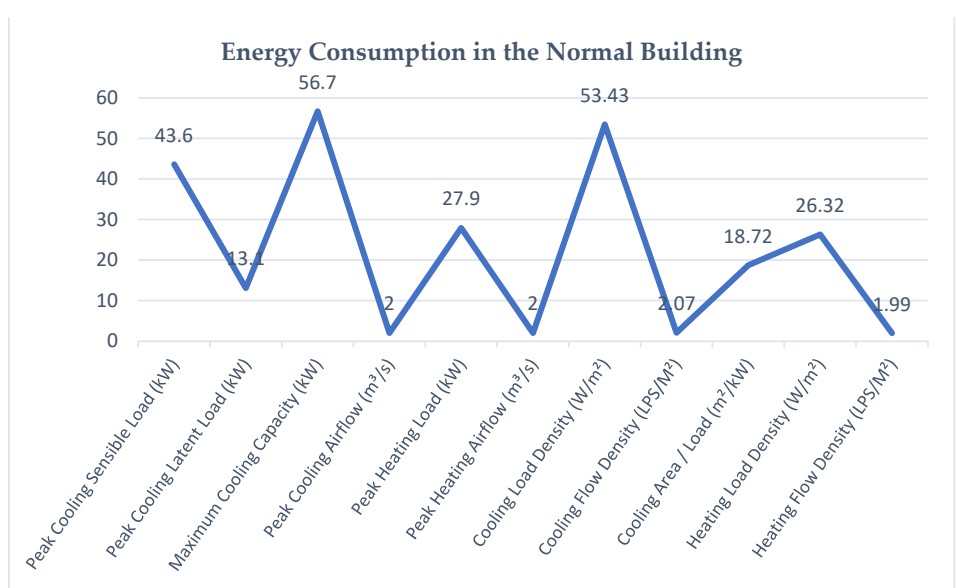

**Figure 8.** Section of the Modeling of the Building with Shavadoon.

Owing to the fact that the traditional residential building used to be a flat or had a maximum of two floors, these days, most of the residential buildings are constructed as high-rises, and installing a duct inside the building is necessary (Figure 9).

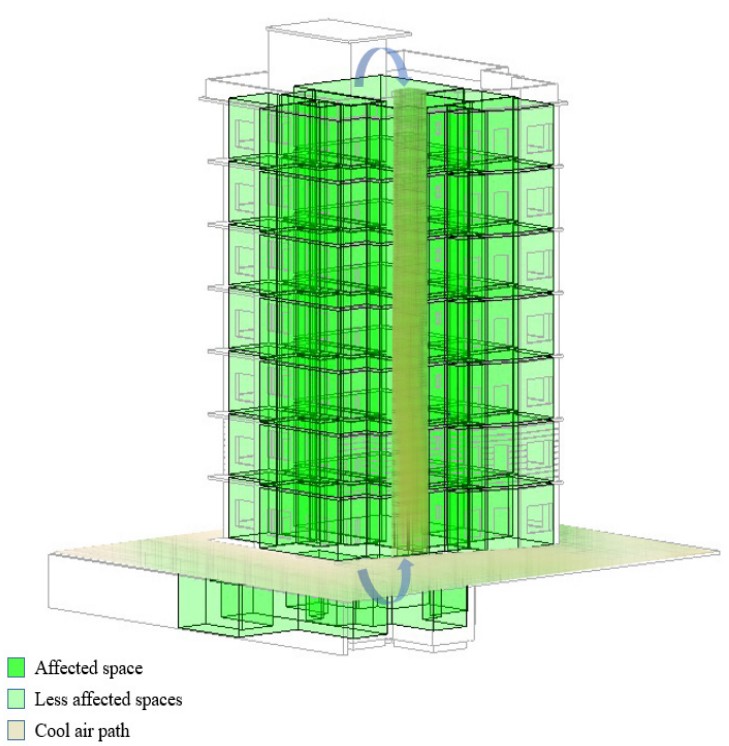

**Figure 9.** Modeling of the Building with Shavadoon (by authors).

Due to the humidity in Lefkoşa, the building has windows in every direction, so the building is extroverted, which has natural ventilation, and this has a positive impact on the energy consumption of the building. A re-analysis was performed, and the obtained statistics showed that the building built in Lefkoşa without the Shavadoon is economical and exhibits low consumption (see Figure 10).

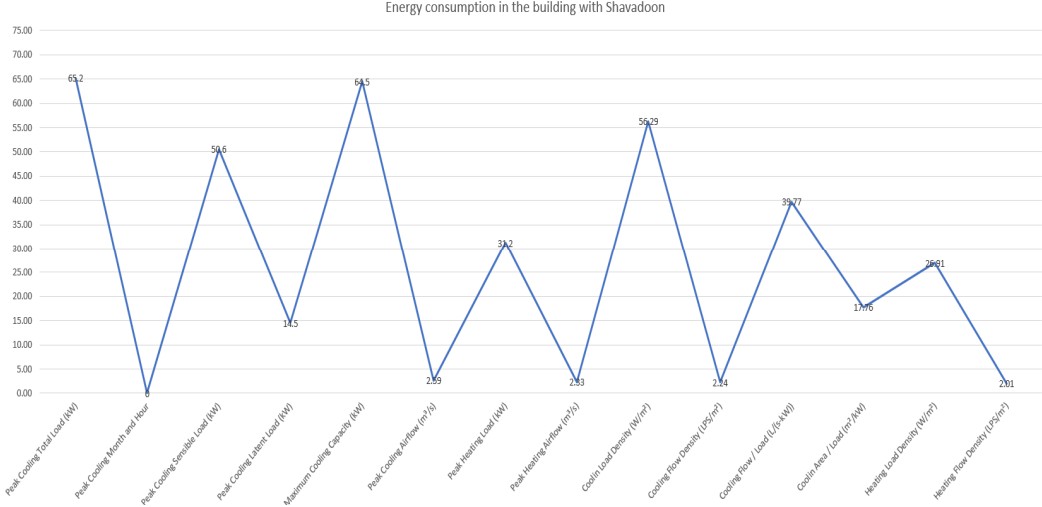

**Figure 10.** Energy consumption in the building with Shavadoon (by authors, 2022).

On the other hand, the buildings in Dezful are introverted, so the analysis is conducted, and the comparison between the two results is demonstrated in Figure 11. Figure 11 demonstrates that the whole building built in Lefkoşa is economical in terms of energy consumption.

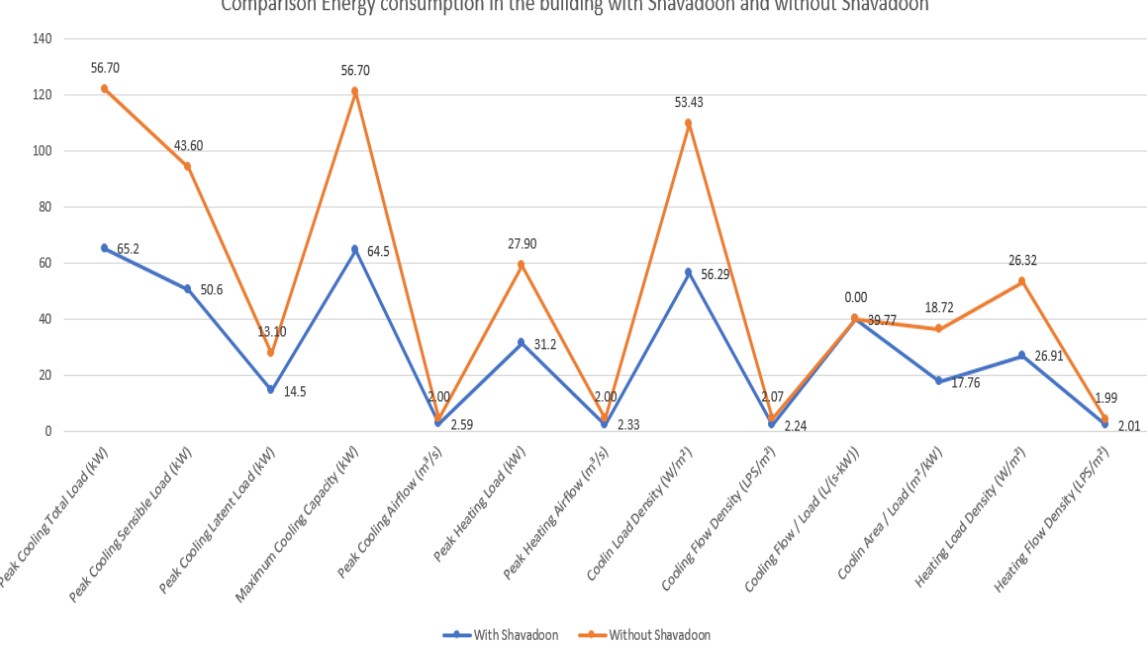

**Figure 11.** Comparison of the energy consumption in the building with Shavadoon and that in the building without Shavadoon in Lefkoşa.

### 5.2. Integration of Shavadoon and Modern Architecture (Results and Findings)

Based on the research conducted on different models of the Shavadoon [27,29], the best performance in terms of thermal comfort is related to the central courtyard and the surrounding Kats, on the basis of which this model has been re-studied in the same climate but in a high-rise building.

The results showed that the external climate and internal building environment are inextricably constrained by each other; thus, a building's energy consumption is closely correlated with outdoor climate factors.

A residential building supplies a convenient indoor environment, and the change in the outdoor climate will affect the building energy consumption. Therefore, similarities in the climate give rise to the point of view that vernacular techniques can be experimented with in instruction to reduce energy consumption at the global level and that spreading this ancient science can help future generations in preserving the environment.

With the addition of the Shavadoon, the cooling area/load of the building is 17.76 (m$^2$/kW), which is higher than the normal situation of 18.72 (m$^2$/kW). So, it can be figured out that, by using the latent potential in the ground, it is possible to cool down the volume of the building, which will have an effect on reducing consumption for cooling in the hot season. In the long term, this technique can have a significant effect on reducing building consumption, since Lefkoşa has 320 sunny days.

On the other hand, duct placement is an effective factor that causes cool airflow from underground to the roof and also combines the cooling air vertical circulation in a high-rise building with the natural ventilation from windows in a horizontal direction. In addition to cooling and reducing energy consumption, it will improve the natural ventilation performance.

### 6. Coalition of the Shavadoon and the Modern Residential Building in North Cyprus (Discussion)

Considering that the Shavadoon has been one of the indigenous techniques in the architecture of Dezful city, it has had various uses.

In addition to being a space for family or neighbor gatherings, it was also responsible for the natural ventilation and cooling of the building in such a way that it had a vent

outside the house for air to enter, and according to the depth and humidity in it, the air in the room was cooled. This cool air is distributed through vents in different parts of the building. This technique can also be used in similar climates to cool spaces, which will reduce energy consumption.

The experiments conducted showed that since the construction styles of the two countries are in two different ways (introverted (Iran) and extroverted (Northern Cyprus)), the residential buildings in Lefkoşa are economical, but with the addition of the Shavadon, the cooling degree of the building is higher. Certainly, this is due to the number of floors. In the ancient architecture, residential buildings were wide and built with a maximum of two floors, so the Shavadoon has had a significant effect on natural ventilation and cooling. However, today, with high-rise construction, cool air from the Shavadoon, which is placed underground, should be used until the last floor by mechanical factors such as ventilators for transferring cool air. On the other hand, Shavadoon has been a place for family gatherings, whereas one of the characteristics of flexible spaces is that with spatial flexibility in a fixed surface area, it is possible to use the Shavadoon space in residential complexes as a multi-purpose space. Due to the natural ventilation and cooling of the space itself, it can be considered the most economical part of modern buildings.

## 7. Conclusions

Recent studies in the world have made the impact of climate change on building consumption clear to everyone; more energy consumption will lead to more pollution and global warming. In this context, it has been estimated that the gradual increase in the average temperature in subsequent years will decrease the demand for thermal energy for heating, but the demand for electricity in the production of cooling will increase.

Since the most important energy consumption in buildings is for heating, cooling, and lighting, the effect of natural ventilation is significant in this field. As a consequence, different buildings have different levels of energy consumption, but the biggest energy consumers are always residential and commercial buildings. Consequently, the most effective element in the building in conserving energy is, after the openings, the frame or the walls. During the cold season, the heat of the building will exist, and in the hot season, the heat will penetrate from the outside into the building. Since insulation and modern techniques are expensive, it is better to deal with the waste of energy with local solutions.

In this regard, being upgraded to be equal to today's living standards is a necessity. By using the characteristics of this space, an attempt was made to investigate the effect of this space on cooling and natural ventilation by simulating this space in the city of Lefkoşa, located in Northern Cyprus, in terms of the cool flow from this space to the building.

Based on the analysis carried out by BIM, it has been determined that humidity plays a significant role. The results illustrated that the building in normal conditions in the city of Lefkoşa is economical in terms of energy consumption. On the other hand, the analyses showed that by adding the Shavadoon to the apartment in Lefkoşa, wide spaces are affected, and this method can help to build mass to be cool.

Therefore, since the underground space was a private space in local architecture, with the change from the private to semi-public usage of the Shavadoon, this space can be classified as "Spatial flexibility in a fixed surface area." Based on the general principles of flexibility in buildings, and with changes in the interior design without changing the overall mass, it can be classified as a polyvalence space, which can be used in the modern apartment where many people live. Furthermore, in addition to influencing the process of reducing consumption, it can also have a positive effect on responding to the requirements of the residents and making the building more sustainable.

Due to the fact that indigenous houses were flat in the past, these days, for various reasons, it has become common to live in an apartment. The Shavadoon is not effective in a native way, which, in these research results, was shown by studying and applying the Shavadoon in relation to contemporary architecture. The integration of air filters and mechanical ventilation in high-rise residential buildings is necessary for stronger

traction and suction in the direction of sustainability, but regarding the debate in zero-energy building design about the balance between energy storage and the distributed consumption of renewable energy (wind power), in the long term, in addition to reducing energy consumption for cooling the building, it will also include a massive reduction in government costs for cooling residential buildings.

Finally, it can be noted that, today, by using BIM, it is possible to examine the fracture between the rich knowledge of the past and the variety of modern techniques and to prevent the trial and errors that have been made, which in turn will lead to a kind of reduction in global consumption.

## 8. Future Research

This research has applied the indigenous model of the Shavadon to a contemporary building in a moderate and humid climate, and it has been investigated in terms of its effect on reducing energy consumption in the field of architecture. This research has the capacity to be investigated in urban planning aspects and the possibility of building underground cities in the direction of sustainability and environmental protection.

**Author Contributions:** H.R. and E.R. collected the data, conceived the study, and developed the methodology; H.R. analyzed and organized the results into figures and tables; E.R. contributed to the manuscript's revision. All authors have read and agreed to the published version of the manuscript.

**Funding:** This research received no external funding.

**Institutional Review Board Statement:** Not applicable.

**Informed Consent Statement:** Not applicable.

**Data Availability Statement:** Not applicable.

**Conflicts of Interest:** The authors declare no conflict of interest.

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
