# Peer review of "The Consequence of Combining Indigenous Techniques with a Flexible Design to Reduce Energy Consumption in Residential Buildings for Future Architecture"

_sustainability, doi:10.3390/su142113958_

Round 1
Reviewer 1 Report
1 - DOI is missing in most of the references (also cossref) and don’t have a list number
2 - All the structure of the test must be reviewed;
3 - Improve the quality of the figures;
4 -Line 44: “Furthermore, there is an important electricity usage affected by humidity.” Doesn’t make sense, please explain better, the meaning of the sentence.
5 - Please replace:
Line 57:
“vernacular solutions met the requairments of residents,” by “vernacular solutions meet the requirements of residents,”
6 - In figure 1, Replace “Case Stady” by Case Study
line 88: Thus, Additional , replace by additional;
7- Please explain in Line 208:
“5–12 m ?” Deep ?
8- Line 218:
The figure 1 ? It’s figure 2, please replace it;
9- Line 231:
“of time is more than 45centigrade. It lets the users to stay away from the scorching heat”
Replace by:
of time is more than 45 centigrade. It lets the users to stay away from the scorching heat 231
10 - Lines 275, 276 e 277:
The temperature must must respect a pattern or appear in kelvin or centigrade;
11- Figure 6 e. 8, must me improved, it’s impossible read the values;
12- Conclusions must explain the benefits of using this “Shavadoon” in the new building, the BIM model doesn’t show the benefits;
13 - future work to be done in energy saving (economical studies must be presented.
Author Response
first of all thanks for all your comments, which I did by yellow highlighted.
Also tried to improve the quality of the figure, and a conclusion is revised.

Reviewer 2 Report
I am not an expert in architecture. The article is devoted to the actual task of reducing the energy consumption of residential buildings. Is the article an overview of solutions?
There are a few questions and comments:
1. Do you use the data for estimating energy consumption in 2013 and 2015? The data is out of date!
2. Figure 8, Figure 11 and Figure 12 are unreadable. In what units and for what period was the power measured?
3. The questions in the article (lines 66-72) do not have an open answer in the studies.
Author Response
first of all thanks for all your comments, which I did by yellow highlighted. Also tried to improve the quality of the figure.

Reviewer 3 Report
Manuscript ID: sustainability-1936964
Review Report
Dear Editors and Authors,
The reviewer would like to submit the reviewed manuscript for the study entitled "Combining Indigenous Techniques with a Flexible Design to Reduce Energy Consumption in Residential Buildings" for consideration in the Sustainability Journal (ISSN 2071-1050).
After the review process, the reviewer would like to give some critical thinking and idea to help authors get their job done efficiently. Comments and Suggestions for authors are the following:
The authors proposed combining indigenous techniques with a flexible design in this study. Much study has focused on energy efficiency and natural ventilation. Fewer studies have examined the global confluence of vernacular and modern architecture. This essay seeks to answer how indigenous strategies for natural ventilation and flexible design might reduce home energy usage. This article examines the Sheva Doon, a solution to escape the excruciating summer heat in Iran, and integrates it with modern architecture in Northern Cyprus to find its effects on cooling the building and reducing energy consumption. This essay will study the impact of local solutions, natural ventilation, flexible design, and modern architecture using BIM. Future architects will be given new ideas for creating energy-efficient homes. In short, the reviewer found that the paper has merits, and it could be acceptable to publish in its present form. Therefore, please revise the manuscript according to the reviewer's comments
1. General comments:
- Title: This paper seems to be a review paper. Please rename the manuscript title.
- Line 4: Please remove the title of the second author.
- Line 6: Please Remove "Ph.D. Candidate.”
- Figures 2, 3, 4, 5, 6, 7, 8, 9, 10, 11, and 12: the figure quality is very poor; please increase the image's resolution so that these figures are more explicit.
- Line 137-141: this passage is too difficult for readers to understand. Please revise
- Lines 43, 88: Please lowercase the words “Cooling,” "Additional," and please rewrite the sentence "Thus, Additional qualitative methods such as grounded theory in the article, BIM will be used to analyze thermic of comfort and energy implementation about natural ventilation."
- Lines 98-100: This sentence is confusing readers. Please Revise.
- Table 1: Format the table according to the instructions in The Journal Template.
- Line 306: Please remove square brackets containing Table 1
- Line 322: Please use plural form for results and finding
- Lines 337, 358, 356: Please remove square brackets containing Figures 7-8 and 9-11. Please use the plural form for Figures 7-8; Figures 9-11
- Line 380: Which figures did you mention?
- Please explain Figures 4-6, 12 in the text. These figures were missed. Please revise.
- There is a shortage of references, and please add more references from Sustainability Journal or others.
- References: Format according to the magazine guidance (template)
- Section 6: Please remove subsection 6.1 because there is only one subsection in this section.
- Where is Section 7 in your manuscript?
- Section 8: Please remove subsection 8.1 because there is only one subsection in this section.
- Please provide pieces of evidence that the authors have been allowed to reuse all figures from other published papers.
- The authors should re-arrange the article logically.
- Please check all English of the whole manuscript.
2. Questions
- What is the most effective process for managing energy consumption in buildings?
- What are the main factors that make the energy management system in the buildings?
- What do the authors think about the new challenge and the global trend of "Future of Houses/buildings - Zero Energy"?
The reviewer hopes that his point of view could help the authors improve their work well.
I appreciate your cooperation.
Sincerely yours,
The Reviewer
Author Response
First of all thanks for all your comments, which I did by yellow highlighted.
for part 2, I read some more articles and added the information which is related to Zero Energy and energy management.

Round 2
Reviewer 1 Report
The references must be improved and the DOI is still missing
Author Response
Dear peer reviewers in the sustainability journal
Thank you for all the comments to improve this article.
Thank you for all the comments to improve this article.
Proceedings were taken to create a DOI since it was necessary to upload this article to get the code, it makes a question mark to questions about the article's specificity and uniqueness. This may damage the credibility of the article. If it is possible please guide me about that.
Therefore, the Orchid code has been received and mentioned in the article.

Reviewer 3 Report
Dear Authors and Editors:
Please provide a point-to-point response letter.
The authors have attached the wrong file (maybe) which is titled the cover letter.
Thank you.
Sincerely yours,
The reviewer.
Author Response
Dear peer reviewers in the sustainability journal
First of all, I would like to thank you for taking the time to read the article and for sharing your expertise and experience. It is so inspiring to have kind and influential reviewers contributing to the development of this article. According to the comments, the following changes have been made and are highlighted in the text:
1_ The previous topic was “Combining Indigenous Techniques with a Flexible Design to Reduce Energy Consumption in Residential Buildings.” According to your comment;
“This paper seems to be a review paper. Please rename the manuscript title.”
The topic has been changed to The Consequence of Combining Indigenous Techniques with a Flexible Design to Reduce Energy Consumption in Residential Buildings for Future architecture.
This article is a combination of a review and a new investigation of the Shavadoon technique. Previously, some researchers had written about natural ventilation in Shavadoon using the software CFD, which gave me the idea to continue this topic by studying the same climate and applying it in high-rise buildings.
2_ In Lines 4 and 6: “Please remove the title of the second author/ Remove "Ph.D. Candidate”. It has been done.
3_ “Please increase the image's resolution so that these figures are more explicit.” Tried to improve resolution in all figures.
4_ “In Lines 98-100: Please revise.” It's revised and shown by highlighting.
5_ “In Line 137-141: Please revise.” Because of adding some paragraphs it shifted to line 146_149 and had been revised.
6_” Table 1: Format the table according to the instructions in The Journal Template.” It has been changed to the correct format in line 325.
7_” Lines 337, 358, 356: Please remove square brackets containing Figures 7-8 and 9-11. Please use the plural form for Figures 7-8; Figures 9-11. “All of them had been corrected.
8_ “The authors should re-arrange the article logically. “Based on this sensitive comment, I attempted to be more careful in the arrangement, writing of the article, and explanations of the figures, and to improve the explanations, which are highlighted in lines 369 _376. Also found in 414_ 436.
9_ “Is the content succinctly described and contextualized with respect to the previous and present theoretical background and empirical research (if applicable) on the topic?” One paragraph has been added to the abstract to make it more clear, as indicated by the red color in the text. The addition of line 54 64 clarifies the significance of the subject. In the previous version, some paragraphs were added in lines 51_ 61. The current theoretical background will be shown in lines 149_149, and the article's empirical part will be shown in lines 367 _374.
10_ “Are all the cited references relevant to the research?” tried to use relevant and updated research.
11_ Due to “Are the research design, questions, hypotheses and methods clearly stated?”
Line 74_ 78 mentions the main question that this article seeks to answer. The methods are also clearly explained in the Materials and Methods section. Furthermore, the article's structure has been slightly altered to improve consistency.
12_“For empirical research, are the results clearly presented?” The obtained results were re-read and for its clarity, it has been tried to be expressed in simpler language.
Last but not the least, I would like to extend my appreciation to respected reviewers who patiently, criticize my manuscript and I hope the revised manuscript has fulfilled the requirements.
Sincerely thanks.

Round 3
Reviewer 3 Report
The authors have corrected the reviewer's comment.
I suggest that the manuscript should be accepted for publication.
Thank you.